# Understanding the Role of Oxidative Stress in Platelet Alterations and Thrombosis Risk among Frail Older Adults

**DOI:** 10.3390/biomedicines12092004

**Published:** 2024-09-03

**Authors:** Diego Arauna, Simón Navarrete, Cecilia Albala, Sergio Wehinger, Rafael Pizarro-Mena, Iván Palomo, Eduardo Fuentes

**Affiliations:** 1Thrombosis Research and Healthy Aging Center, Department of Clinical Biochemistry and Immunohematology, Interuniversity Center for Healthy Aging (CIES), Interuniversity Network of Healthy Aging in Latin America and Caribbean (RIES-LAC), Faculty of Health Sciences, Universidad de Talca, Talca 3460000, Chile; darauna@utalca.cl (D.A.); snunez@utalca.cl (S.W.); 2Unidad de Nutrición Pública, Instituto de Nutrición y Tecnología de los Alimentos, Interuniversity Center for Healthy Aging, Universidad de Chile, Santiago 7810000, Chile; calbala@uchile.cl; 3Facultad de Odontología y Ciencias de la Rehabilitación, Universidad San Sebastián, Sede Los Leones, Santiago 7500000, Chile; rafael.pizarro@uss.cl; 4Interuniversity Network of Healthy Aging in Latin America and Caribbean (RIES-LAC), Santiago 7810000, Chile

**Keywords:** oxidative stress, frailty, platelets, cardiovascular disease, thrombosis

## Abstract

Frailty and cardiovascular diseases are increasingly prevalent in aging populations, sharing common pathological mechanisms, such as oxidative stress. The evidence shows that these factors predispose frail individuals to cardiovascular diseases but also increase the risk of thrombosis. Considering this background, this review aims to explore advances regarding the relationship between oxidative stress, platelet alterations, and cardiovascular diseases in frailty, examining the role of reactive oxygen species overproduction in platelet activation and thrombosis. The current evidence shows a bidirectional relationship between frailty and cardiovascular diseases, emphasizing how frailty not only predisposes individuals to cardiovascular diseases but also accelerates disease progression through oxidative damage and increased platelet function. Thus, oxidative stress is the central axis in the increase in platelet activation and secretion and the inadequate response to acetylsalicylic acid observed in frail people by mitochondrial mechanisms. Also, key biomarkers of oxidative stress, such as isoprostanes and derivate reactive oxygen metabolites, can be optimal predictors of cardiovascular risk and potential targets for therapeutic intervention. The potential of antioxidant therapies in mitigating oxidative stress and improving cardiovascular clinical outcomes such as platelet function is promising in frailty, although further research is necessary to establish the efficacy of these therapies. Understanding these mechanisms could prove essential in improving the health and quality of life of an aging population faced with the dual burden of frailty and cardiovascular diseases.

## 1. Introduction

By 2050, the number of older adults worldwide will double (from 11% to 22%), which could be accompanied by a significant increase in geriatric syndromes such as frailty syndrome [1,2,3]. Frailty is a relevant geriatric syndrome in older adults associated with unhealthy aging, present in 25–50% of individuals with cardiovascular diseases (CVDs), reaching up to 70–80% in conditions such as heart failure or valvular aortic disease [4,5]. This syndrome has been acknowledged as a potential prognostic factor for coronary disease and is associated with a 2.5–3.5-fold increased risk of mortality, even in patients with less severe cardiovascular disease [4,6]. The evidence underscores a bidirectional relationship between frailty and CVDs, elucidating intricate vascular, cardiac, and muscular alterations [5]. Similarly, advanced age (75 years or older), a well-recognized non-modifiable risk factor for CVDs, is also associated with a higher risk of developing frailty [5]. Notably, recent investigations at the molecular level have shown that both frailty and CVDs exhibit shared alterations concomitant with procoagulant and pro-inflammatory states [6,7]. There are different definitions and scales associated with frailty, one of the standards being the definition proposed by Fried et al. based on five physical criteria, (i) loss of strength, (ii) increase in fatigue, (iii) decrease in walking speed, (iv) low physical activity, and (v) involuntary weight loss, which have been associated with alterations in the redox state and oxidative stress [8]. Thus, the accumulated evidence shows that frailty syndrome has a robust oxidative stress component, both at the circulating and cellular level, affecting different antioxidant defenses, such as inductors of reactive oxygen species (ROS) overproduction [9,10,11,12,13]. In addition to this, various studies have proposed that oxidative stress biomarkers such as isoprostanes, carbonylated proteins, malondialdehyde, and lipoprotein-associated phospholipase A2 (Lp-PLA2) could predict the progression or development of frailty [11,12,14,15]. The heightened oxidative stress observed in frail patients has been attributed to a depletion of enzymatic and non-enzymatic antioxidant defenses [12,16,17]. Nevertheless, the specific components primarily affected remain unclear [12]. It is yet to be elucidated whether this decline in antioxidant defenses results from alterations in protein expression levels or the inhibition of their activity [18,19].

One of the primary causes of mortality in frail older individuals is represented by CVDs; oxidative stress is recognized as a pivotal factor in their development [6]. Within the pathophysiological mechanisms of CVDs, alterations in platelet functionality, and their interactions with other cells relevant to atherothrombotic processes, play a crucial role [20]. In these alterations, damage induced by oxidative stress and mitochondrial dysfunction may potentially contribute to the formation of hyperactivated platelets and increased platelet functionality, phenomena also reported in frail patients [14,21]. Thus, oxidative stress seems to be a transversal axis between frailty syndrome and CVDs in the context of aging. Considering this context, this review aims to describe the advances in the role of oxidative stress in platelet alterations and CVDs associated with frailty. Through this, we address the research question of how these factors contribute to the progression of CVDs in frail older adults.

### 1.1. Platelet Alterations in Frailty Syndrome: Role of Oxidative Stress

During platelet activation, a significant shift occurs in both the redox balance and the mitochondrial metabolism of platelets [22]. Various signaling pathways have been identified that lead to the production of ROS by Nicotinamide Adenine Dinucleotide Phosphate (NADPH) oxidase (NOX) and mitochondria in full platelet activation; nevertheless, this overproduction of ROS may not be indispensable for maintaining primary hemostasis and could potentially contribute to the augmentation of platelet functionality [23]. The overproduction of ROS in platelets and their release at the vascular level contribute to activating other platelets and processes of adhesion and recruitment, creating a self-amplifying cycle [15]. This cyclic process ultimately results in a procoagulant platelet phenotype, heightened reactivity, and increased apoptosis, thereby contributing to an elevated risk of thrombosis in CVDs and unhealthy aging [14,15].

In the year 2015, Shi et al. analyzed the expression of glycoprotein (GP) IIb/IIIa on the platelet surface (activation marker, PAC-1), the formation of platelet and monocyte aggregates (PMAs), and the plasma level of the chemokine CCL5 (RANTES) against a weak agonist stimulus (adenosine diphosphate, ADP, 1 µM). They observed that under an unstimulated condition (control), there was no difference in activated GP IIb/IIIa expression and RANTES levels between the groups studied; however, in the stimulus condition with ADP 1 µM, the older frail group showed significantly higher increases in GP IIb/IIIa activation compared to the control groups (healthy young and healthy older adults) [24]. The interaction between fibrinogen and GP IIb/IIIa activation induces platelet aggregation, showing that ROS can regulate the affinity and activation of this receptor by interacting with the thiol groups in its protein structure [15,25,26]. Likewise, for GP IIb/IIIa activation, the activation of inositol tri-phosphate kinase (PI3K) and phospholipase C is essential, as they are involved in collagen-induced ROS generation, NADPH oxidase activation, superoxide production, and platelet aggregation [15,27]. It has been described that the aging process and frailty could increase the plasma fibrinogen level, independent of chronic diseases [28,29]. This context shows an increase in the ligand of the GP IIb/IIIa (fibrinogen) in a context of high oxidative stress, which could be part of the mechanistic explanation of the increase in GP IIb/IIIa activation in frail older people and of the increase in thrombotic events in frail older adults [30,31].

The increase in oxidative stress in patients with CVDs, both at the circulatory and intracellular level, has been suggested as a possible cause of low response to acetylsalicylic acid (ASA) [32]. In 2016, Nguyen et al. detected that frail older adults who consume aspirin have higher adjusted arachidonic acid agonist test (ASPI) measures, suggesting a reduced response to aspirin compared with non-frail older adults [33]. It has been reported that ASA resistance in coronary artery disease, pulmonary disease, metabolic syndrome, cancer, and other chronic diseases is associated with an increase [21,32]. It has been proposed that the forming of isoprostanes could have a crucial role in ASA resistance mediated by oxidative stress [34]; a similar situation could be observed in frail patients, where an increase in 8-isoprostane at the plasma level (a biomarker of oxidative stress) has been reported [35].

In 2019, Hernández et al. showed that frailty syndrome would be associated with an increase in platelet activation (shown in an increase in P-selectin expression) compared to non-frail older adults. An increased expression of P-selectin is associated with a more significant mobilization of platelet α granules, depending on an increase in intracellular calcium and ROS generation [15,36]. It has been reported that the rise in circulating oxidative stress present in frail older adults could induce an increase in intracellular calcium release from blood cells (including platelets); however, this has not been clarified [37]. So far, it has been reported that platelets stored for transfusion have shown a positive correlation between an increase in ROS and the expression of P-selectin, a situation that can occur in the platelets of frail patients [38].

In the year 2022, Arauna et al. presented a noteworthy study revealing that frail elderly patients, classified according to the Fried phenotype, exhibited heightened platelet activation (evidenced by the exposure of PAC-1 and P-selectin) and aggregation in response to both aggregating and sub-aggregating stimuli of ADP, a physiologically relevant agonist [35]. The observed platelet hyperactivity was concomitant with an elevation in plasma biomarkers associated with platelet activation (thromboxane B2), oxidative stress (8-isoprostane), and mitochondrial dysfunction (Growth Differentiation Factor 15, GDF-15) [35]. The coherence of this evidence underscores a potential state of platelet hyper-reactivity in frail patients, with oxidative stress secondary to mitochondrial dysfunction potentially playing a pivotal role in its origin and progression [35,39]. The main findings about platelet alterations in frail individuals are summarized in Table 1.

On the other hand, in the year 2023, Arauna et al. reported that frail individuals diagnosed using the Frail Trait Scale 5 (FTS-5) tool exhibited an increase in plasma levels of procoagulant microvesicles derived from platelets [7]. This finding is associated with the previously reported increase in platelet activation. It is known that extracellular vesicles, including microvesicles, have a bidirectional relationship with the redox state and ROS production, transporting both antioxidant and oxidant molecules and altering intracellular ROS levels [40,41]. Additionally, an increase in microvesicle production is linked to redox state alterations and ROS overproduction, acting as a response factor to cellular damage [42]. It is worth noting that circulating microvesicles have been proposed as biomarkers for CVDs and are helpful in risk stratification and clinical decision-making [43,44].

At the vascular level, oxidative stress induces various dysregulations in endothelial cells, leading to their activation and subsequent dysfunction [45,46]. Endothelial dysfunction, in turn, results in the exposure of platelet adhesion molecules and a reduction in various regulatory molecules for platelet function, such as nitric oxide (NO) [47]. Despite the documented presence of endothelial dysfunction in frail elderly individuals, its precise role in the observed platelet hyperactivity within this syndrome, the underlying mechanisms, and how these alterations might impact the platelet–endothelium interaction remain unclear. Further research is warranted to elucidate these intricate aspects of pathophysiology in frailty-related platelet dysfunction and vascular complications [3,5].

Frailty has been associated with endothelial dysfunction, as evidenced by the following serum markers: intercellular adhesion molecule 1 (ICAM-1), endothelin 1 (ET-1), von Willebrand factor (vWF), plasma thrombomodulin, and asymmetric dimethylarginine (ADMA) [48,49,50,51,52]. ICAM-1 and vWF are proteins that facilitate the process of platelet adhesion, whereas ET-1 could influence platelet functionality in patients with coronary artery disease or myocardial infarction [53]. In frail older individuals, an elevation in some of these markers is observed, such as the level of ADMA [52]. ADMA is an inhibitor of nitric oxide synthase (NOS), associated with oxidative damage, and is an independent cardiovascular risk factor. The decrease in its activity reduces NO production and diminishes the capacity for vascular anti-aggregation [54,55].

### 1.2. Oxidative Stress: A “Bridge” between Frailty and Cardiovascular Diseases

#### 1.2.1. Frailty and CVDs

It is well known that aging is a risk factor for CVDs, and this risk is even more pronounced in an unhealthy aging process. The altered cellular processes behind aging, which induce the increased risk of CVDs, are not fully elucidated yet. However, recent advances indicate that it is a multifactorial process. Among the most notable processes, telomere shortening, chronic low-grade inflammation, oxidative stress, mitochondrial dysfunction, the accumulation of senescent cells, and reduced autophagy are highlighted [56]. Additionally, these alterations play a crucial role in components such as disability, decreased physical activity, fatigue, and weight loss [56]. Also, it is well known that oxidative stress can cause, or be a consequence of, various CVDs and metabolic diseases, making it essential to clearly distinguish whether the condition is induced by oxidative stress or vice versa. In this regard, frailty appears as a factor that could exacerbate ROS overproduction and the depletion of antioxidant capacity in CVDs [13,35,57].

Recent studies indicate that frailty syndrome and CVDs have a bidirectional relationship, which is not fully elucidated. Older individuals with cardiovascular risk factors, a subclinical CVD, or a diagnosed CVD have a higher risk and prevalence of frailty or pre-frailty [4,58,59,60,61]. Veronese et al., in a study that incorporates extensive frailty measurement in a cohort comprising 4211 older individuals and with an average follow-up of 8 years, found that frailty syndrome increased the risk of CVDs, reinforcing the notion of the predictive capacity of this syndrome [62]. Also, frailty has a significant social component, with social isolation, depression, and a decline in the support network of family or friends increasing the risk of developing this syndrome [3]. Likewise, it has been observed that these social aspects are also associated with a higher risk of developing CVDs [3].

#### 1.2.2. Oxidative Stress in CVDs and Frailty

As described earlier, the central common pathways between frailty syndrome and CVDs are endothelial dysfunction, inflammation, and high levels of oxidative stress [13,52,63]. It has been reported that frail older adults present high levels of oxidative stress, which is associated with a high risk of CVDs [13,64]. The relationship between frailty, oxidative stress, and CVDs is not limited to vascular damage but also involves alterations at the muscular, adipose tissue, and bone tissue levels [59]. A cardiac-level frailty is present in 70–80% of conditions, such as heart failure or valvular aortic disease [4,5]. In this relationship, oxidative stress plays a central role due to its involvement in molecular modifications that disrupt the homeostasis of cardiac tissue and associated blood vessels, such as the oxidation of lipoproteins, reduction in nitric oxide (NO) bioavailability, activation of pro-inflammatory signaling pathways like NF-κB, upregulation of adhesion molecules (e.g., VCAM-1, ICAM-1), release of cytokines (e.g., TNF-α, IL-6), and fibrosis [65]. Also, frail older adults present left ventricular hypertrophy, diastolic dysfunction, reduced left ventricular ejection fraction, and increased arterial stiffness [66,67]. In this context, oxidative stress can contribute to developing these cardiac changes and induce a higher risk of heart failure and other cardiovascular complications in this population [68].

At the muscular level, frailty and CVDs share the development of sarcopenia, which is understood as a process involving the loss of quantity and quality of muscle fibers, impacting physical activity and various indicators of cardiovascular disease risk. Some reports suggest that dietary supplementation with antioxidants such as vitamin E, vitamin C, carotenoids, and resveratrol may be a potential therapeutic intervention in sarcopenia [69]. However, the current evidence is not clear about the therapeutic efficiency [70]. At the bone level, frailty and increased oxidative stress are associated with a higher risk of fractures and falls, which has also been linked to an increased risk of developing CVDs and thrombosis. At the adipose tissue level, frailty is associated with obesity, a well-recognized cardiovascular risk factor, indicating an increased accumulation of this tissue. On the other hand, recent advances suggest that oxidative stress also plays a crucial role in hearing loss and the development of cognitive diseases such as Alzheimer’s in frail individuals. This evidence suggests a cross-cutting impact of oxidative stress in the development of frailty and its associated factors [71].

A study by Saum et al. in older adults in Germany (ESTHER cohort) tested different markers of oxidative stress, such as the biological antioxidant potential (BAP), derivate reactive oxygen metabolites (d-ROM), and the total thiol level (TTL). The results corroborated high levels of circulating d-ROM (2-fold compared to the non-frail group) and the loss of about 50% of BAP and TTL capacity [19]. High levels of d-ROM are associated with thrombotic events in patients with CVDs [72].

In a recent literature review by Sepulveda et al., evidence on oxidative stress biomarkers evaluated in frail older individuals was compiled [73]. In this context, various studies indicate that frailty syndrome is primarily associated with the generation of isoprostanes, Lp-PLA2, and homocysteine levels. In contrast, other biomarkers, such as osteoprotegerin, are associated with decreased walking speed, a criterion for frailty, and an indicator of CVD risk [73]. Additionally, the association of the 8-hydroxy-2-deoxyguanosine is highlighted, one of the most widely recognized biomarkers of oxidative DNA damage [73]. All the factors above are also associated with CVDs, disability, and a high risk of death [74].

Among the elements that regulate ROS production, the prominent organelle in this function is the mitochondria through oxidative phosphorylation and its complexes I and III [75,76]. A lower mitochondrial DNA (mtDNA) copy number was associated with frailty [77]; however, this has not been corroborated by studies of mitochondrial functionality in frail, pre-frail, and robust groups. High levels of ROS produce alterations in calcium metabolism and mitochondrial dysfunction [78], which are essential components in the pathophysiology of CVDs [79]. In this respect, the following alterations have been observed in frail older adults: (i) mitochondrial dysfunction in muscle tissue [80], which is related to the sarcopenia present in frail patients; (ii) an increase in malondialdehyde (MDA) levels and oxidized glutathione, which are related to thrombotic events (40 times higher in patients 24 h post thrombosis) [16,81]; and (iii) reduction in non-enzymatic antioxidant parameters [18,19], which is a protector factor against CVDs and thrombosis [82]. All these related changes cause a greater predisposition to thrombotic events [83,84], in which it has been described that platelets play an essential role in thrombus formation—both its development and progression [14,85].

The exact causes of the overproduction of ROS observed in frail individuals are not yet fully clarified. Current evidence suggests that an increase in ROS generation from mitochondria and NADPH oxidase and a decrease in antioxidant defense mechanisms could act simultaneously [9,12,17,86,87]. In this regard, a reduction in the levels of natural dietary antioxidants such as lycopene, lutein/zeaxanthin, and β-cryptoxanthin, as well as ophthalmic acid, has been observed in frail individuals [88]. However, given the variability in the diagnostic tools used, it is still unclear which enzymatic and non-enzymatic antioxidant systems might be associated with frailty [12,13,89]. On the other hand, inflammaging, defined as a low-grade inflammation aging-associated condition, is considered one of the main risk factors related to the development of CVDs [90]. This process causes NLRP3 inflammasome activation, leading to decreased antioxidant response and mitochondrial dysfunction, which are also altered in CVDs [90].

It is essential to highlight that frailty intervention has been evaluated as a potential therapy for preventing and treating CVDs [3]. The evidence suggests that multi-component interventions would represent the most beneficial strategy for frail older individuals with CVDs [3]. However, it is still unclear which specific therapies or domains of frailty would be the most effective to address [3]. At the clinical level, melatonin, proprotein convertase subtilisin/kexin type 9 inhibitors, carvedilol, and metformin have been the main therapeutic drugs with antioxidant effects showing therapeutic or preventive effects in CVDs, particularly in heart failure [91]. In this regard, recent reports have indicated that supplementation with the antioxidant vitamin E through dietary enrichment represents a promising strategy for reducing cardiovascular risk [92]. On the other hand, oxidative stress biomarkers such as NADPH oxidases, advanced glycation end-products, and myeloperoxidase are currently considered promising in the diagnosis and risk stratification of cardiovascular diseases, including heart failure [93,94]. Furthermore, incorporating proteomic studies, endothelial function assessments, and vascular functional studies represents an innovative strategy for evaluating the risk and prognosis of CVDs [94].

### 1.3. Mitochondrial Dysfunction as a Link between Frailty and Thrombosis

Dysregulated mitochondrial metabolism is described as a potential root cause of age-related frailty, with observed alterations in mitochondrial processes associated with the metabolism of vitamin E and carnitine [87]. Mitochondrial dysfunction is generally defined as a decrease in energy production through adenosine triphosphate (ATP) and an increase in ROS production [95,96]. Additionally, it is characterized by alterations in the synthesis and degradation of proteins and lipids, as well as playing a critical role in apoptosis, being relevant in different CVDs, such as heart disease [97,98]. On the other hand, mitochondrial dysfunction plays a crucial role in calcium homeostasis as a critical ion in platelet and cardiovascular functions [96,99,100]. Thus, mitochondrial dysfunction has been closely associated with atherothrombotic processes, heart disease, hypertension, and acute myocardial infarction, among others, recently emerging as a potential therapeutic target in these pathologies [97]. Frailty syndrome has been associated with an increased risk of thrombosis and has been detected more in patients with CVDs [101]. As described earlier, frailty syndrome is present in 25–50% of patients with a high prevalence of classic cardiovascular risk factors (CVRFs), such as hypertension, obesity, dyslipidemia, and diabetes, which individually or collectively create a prothrombotic environment [102,103]. Considering this evidence and that previously mentioned, it is clear that thrombosis is a CVD that occurs more often in frail elderly people [31,104].

This pathology is a chronic inflammatory process of the arterial wall started by endothelial dysfunction, which potentiates the activation and migration of different immune cells and platelets [20,105,106,107]. In the initial stage, the platelets adhere to the damaged endothelium, secreting and exposing molecules that amplify the inflammatory process [20,108], while in the final stage, after the plaque breaks, the platelets adhere and form a thrombus [20,109].

Throughout this process, mitochondrial dysfunction and the increase in oxidative stress represent cellular mechanisms of great relevance, both at the endothelial and platelet levels [96,99]. It has been reported that various agents regulating mitochondrial function exert antiplatelet effects, reducing the risk of thrombosis in murine models [110]. Platelets contain 2 to 5 mitochondria [111]; therefore, integrity is crucial to platelet functionality [112]. It has been described that platelet mitochondrial dysfunction increases intramitochondrial ROS, leading to platelet hyperactivation [110,113]. This platelet mitochondrial dysfunction has been described in CVDs and CVRF, such as pulmonary hypertension [114], diabetes [115], and aging [110]. In one of these studies, a murine aging model was used, and it was found that the platelet mitochondria present increased ROS levels, the depolarization of the mitochondrial inner membrane, the generation of mitochondrial permeability transition pore, and cytochrome c release [110]. These mitochondrial changes were associated with increased hydrogen H_2_O_2_, fibrinogen binding, and GP IIb/IIIa activation, which were also evidenced in frail persons [24].

On the other hand, Fuentes et al. indicate that mitochondrial dysfunction in platelets has been correlated with hyper-reactivity and apoptosis [99,116]. This condition has been identified as one of the therapeutic targets for the prevention and treatment of CVDs in frail older individuals. [99]. Likewise, it is emphasized that developing pharmacological or nutraceutical tools that regulate oxidative stress and prevent mitochondrial dysfunction in the platelets of frail individuals could be a valuable tool in thrombosis prevention in these patients [99]. The platelet abnormalities described in frail patients regarding the response to ASA and increased platelet aggregation and activation are believed to be primarily associated with the elevation of circulating oxidative stress and the impact of oxidative stress on the vascular endothelium. However, the underlying cellular mechanisms altered in platelets are still unclear [33]. In this regard, a study has identified the metabolic signature associated with resistance to ASA in platelets, revealing an overproduction of ROS and alterations in mitochondrial function [32]. These phenomena could also occur in frailty syndrome [117]. In addition to the above, different studies have shown that a high level of procoagulant platelets may cause a low response to antiplatelet therapy [118]. This aspect has not been fully clarified in frail older individuals and could be part of the pathophysiological mechanism underlying the low response to ASA [33]. Current evidence suggests that the generation of procoagulant platelets is a process dependent on the production of ROS, mitochondrial dysfunction, and apoptosis. In this process, there is an increase in the exposure of phosphatidylserine in platelets and the release of other procoagulant factors [118]. Likewise, it has been reported that the platelet mitochondrial dysfunction and protein kinase C signaling pathway could participate in the mechanisms associated with low therapeutic response to antiplatelets, altering intracellular Ca^2+^ levels, ROS production, and ATP production [99,119,120].

Finally, in frail patients with diabetes and hypertension, it has been demonstrated that the use of empagliflozin, a sodium-glucose cotransporter 2 inhibitor, improves parameters associated with frailty, both at cognitive and physical levels, through a mitochondria-dependent mechanism [121]. Furthermore, recent advances in murine models indicate that mtDNA alteration is sufficient to generate various components of frailty, such as sarcopenia and oxidative stress, simultaneously leading to the development of CVDs, such as pulmonary hypertension and heart failure [122]. However, it is necessary to confirm the decline in mitochondrial functionality in frail individuals, along with the increase in cardiovascular risk and the deterioration of cardiac function, as well as the molecular mechanisms involved in identifying therapeutic targets. A summary of the relationship between oxidative stress, cardiovascular diseases, and frailty syndrome can be observed in Figure 1.

## 2. Conclusions

CVDs and frailty have a close bidirectional relationship, producing a systemic increase in ROS production and adverse effects on antioxidant systems, leading to increased circulating and cellular oxidative stress, as evidenced by different biomarkers. The pro-oxidative state reported in frail persons is generated by an overproduction of ROS and the depletion/downregulation of antioxidant enzymes, a situation that can occur in frail people and is part of the mechanism behind increased platelet function, low response to ASA, and increased risk of thrombosis associated with frailty. Thus, the current evidence shows that oxidative stress, with a possible mitochondrial dysfunction mechanism, could be the central axis in the increase in platelet activation and secretion observed in frail people and the low response to ASA. However, more studies are needed to confirm these possible mechanisms in platelets from frail people and in CVDs. Finally, this review elucidates the pivotal role of oxidative stress in the interplay between frailty and CVDs, highlighting its impact on platelet dysfunction and the potential for targeted therapeutic strategies. Also, it underscores the need for further research to validate the molecular and cellular mechanism implicated, identifying new potential therapeutic targets in frail individuals with CVDs.

## Figures and Tables

**Figure 1 biomedicines-12-02004-f001:**
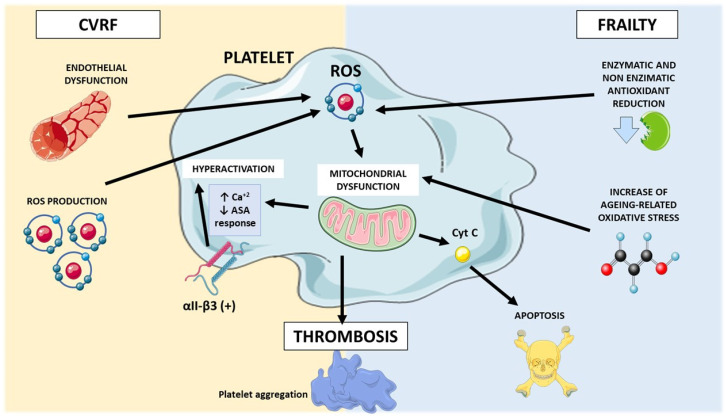
Relation between oxidative stress, cardiovascular diseases, and frailty syndrome. CVRFs, cardiovascular risk factors; Cyt, cytochrome C; ROS, reactive oxygen species; ASA, acetylsalicylic acid; αII-β3, GP IIb/IIIa.

**Table 1 biomedicines-12-02004-t001:** Main findings on platelet alterations in frail older adults.

Frailty Tool	Methodology	Results	Reference
Fried phenotype	Flow cytometry	Frail individuals showed a higher GP IIb/IIIa activation in platelets stimulated by ADP 1 µM	[24]
Edmonton Frail Scale	ASPI	Frail individuals showed a reduced response to ASA in platelets compared to non-frail individuals	[33]
Fried phenotype	Flow cytometry and aggregometry	Frail individuals showed increased P-selectin expression in platelets stimulated by TRAP-6 2.5 µM	[39]
Fried phenotype	Flow cytometry and aggregometry	Frail individuals showed higher GP IIb/IIIa activation and increased P-selectin expression in platelets stimulated by ADP 0.5 µM	[35]

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
