# Peer review of "Understanding the Role of Oxidative Stress in Platelet Alterations and Thrombosis Risk among Frail Older Adults"

_biomedicines, 2024, doi:10.3390/biomedicines12092004_

Round 1

Reviewer 1 Report

Comments and Suggestions for Authors

The review manuscript of Arauna et al. is on the platelet alterations induced by oxidative stress in frail people, which may explain high CVD prevalence among frail people.

They collect relevant literatures in the related fields and put them together in well organized manner in this manuscript. Thus, this manuscript is informative and insightful for the understanding of oxidative stress, frailty, and CVD.

The followings are comments.

1.      Now it’s widely accepted that oxidative stress is produced by and/or it induces various pathological conditions. It is not an overstatement that oxidative stress is associated with all of the pathological conditions as a cause and/or result. In this respect, it is easy to connect any conditions by mediating oxidative stress. Thus, it is important to clearly distinguish if the condition is induced by oxidative stress or if the oxidative stress is produced by the condition.

2.      Disturbance of social relationship is an important, essential part of frailty. The authors deal with only physical aspect of frailty in relation to oxidative stress in this manuscript: it seems to me that it is possible that the theme of this review is on oxidative stress and CVD in “old” persons. How do the authors justify their focus on frailty?

Author Response

Reviewer #1

The review manuscript of Arauna et al. is on the platelet alterations induced by oxidative stress in frail people, which may explain high CVD prevalence among frail people.

They collect relevant literatures in the related fields and put them together in well organized manner in this manuscript. Thus, this manuscript is informative and insightful for the understanding of oxidative stress, frailty, and CVD.

The followings are comments.

  1. Now it’s widely accepted that oxidative stress is produced by and/or it induces various pathological conditions. It is not an overstatement that oxidative stress is associated with all of the pathological conditions as a cause and/or result. In this respect, it is easy to connect any conditions by mediating oxidative stress. Thus, it is important to clearly distinguish if the condition is induced by oxidative stress or if the oxidative stress is produced by the condition.

R: We appreciate the reviewer's suggestion and agree with that perspective. Both frailty and CVD have a bidirectional relationship with oxidative stress. However, recent evidence indicates that frailty could exacerbate both the overproduction of ROS and the depletion of antioxidant defenses in the subclinical stages of CVD, promoting their development. This point will be added to the section “Oxidative stress: A 'bridge' between frailty and cardiovascular diseases.” It can be observed in lines 188-192 in red font.

  1. Disturbance of social relationship is an important, essential part of frailty. The authors deal with only physical aspect of frailty in relation to oxidative stress in this manuscript: it seems to me that it is possible that the theme of this review is on oxidative stress and CVD in “old” persons. How do the authors justify their focus on frailty?

R: We appreciate the reviewer's suggestion and correction. As indicated, frailty syndrome has an important social component, and it has been observed that social isolation, depression, and a deterioration of the family and/or friends support network increase the risk of developing this syndrome. Likewise, it has been observed that these social aspects associated with a higher risk of frailty have also been associated with a higher risk of developing CVD. This clarification was added in the section “Oxidative stress: A 'bridge' between frailty and cardiovascular diseases.”, and can be observed in lines 199-202 in red font. The approach used in this review is focused on older adults because frailty is a geriatric syndrome that primarily affects individuals aged 60 and over. Additionally, age is one of the main non-modifiable cardiovascular risk factors, making the promotion of healthy aging and the reduction of frailty levels promising strategies for the prevention and treatment of CVD. In this context, platelet alterations and oxidative stress emerge as important mediators. Considering the reviewer's suggestion, this clarification was added to the 'Introduction' section and can be observed in lines 45-47 and 50-53 in red font.

Reviewer 2 Report

Comments and Suggestions for Authors

The manuscript titled "Understanding the Role of Oxidative Stress in Platelet Alterations and Thrombosis Risk Among Frail Older Adults" explores the intricate relationship between oxidative stress, platelet dysfunction, and cardiovascular diseases (CVD) in frail elderly populations. While the manuscript provides a comprehensive review of the current literature, several areas require attention and improvement to enhance the clarity and impact of the work.

1.     The objectives and rationale of the study are not clearly stated at the beginning of the manuscript. A more explicit statement of the research questions or hypotheses would provide better context for readers.

2.     While the manuscript includes a substantial review of existing literature, some sections are overly descriptive without sufficient critical analysis.

3.     The manuscript’s structure can be improved by better organizing sections to flow logically from one to the next. The current organization sometimes makes it challenging to follow the progression of ideas. For example, separating the discussion of oxidative stress, platelet alterations, and their implications for CVD into distinct sections with clear subheadings could improve readability.

4.     The manuscript could benefit from additional figures or tables summarizing key findings from the literature.

5.     The analysis of how oxidative stress biomarkers can predict cardiovascular risk and the potential of antioxidant therapies lacks depth.

6.     Several sections of the manuscript contain grammatical errors and awkward phrasings that detract from the overall readability. For instance, the term "pro-thrombotic state" is used inconsistently.

7.     Highlighting how this review advances understanding in the field or identifies new research directions would add significant value.

8.     Include more details on the methodologies of the reviewed studies to provide a clearer picture of how conclusions were reached.

9.     Add summary tables of key studies, their findings, and proposed mechanisms, as well as figures illustrating the relationships between oxidative stress, platelet dysfunction, and CVD.

Comments on the Quality of English Language

Several sections of the manuscript contain grammatical errors and awkward phrasings that detract from the overall readability.

Author Response

Reviewer #2

The manuscript titled "Understanding the Role of Oxidative Stress in Platelet Alterations and Thrombosis Risk Among Frail Older Adults" explores the intricate relationship between oxidative stress, platelet dysfunction, and cardiovascular diseases (CVD) in frail elderly populations. While the manuscript provides a comprehensive review of the current literature, several areas require attention and improvement to enhance the clarity and impact of the work.

  1. The objectives and rationale of the study are not clearly stated at the beginning of the manuscript. A more explicit statement of the research questions or hypotheses would provide better context for readers.

R: We appreciate the correction indicated by the reviewer. In consideration of this, we have added the research question and objective as an explicit statement. This can be observed in lines 79-82 in red font.

  1. While the manuscript includes a substantial review of existing literature, some sections are overly descriptive without sufficient critical analysis.

R: Considering the reviewer’s correction, a critical analysis of the relationship between oxidative stress and platelet alterations and heart diseases was added, as well as how oxidative stress can be both a cause and a consequence of cardiovascular diseases and the studies needed to confirm the described findings. Additionally, the social component of frailty as a factor associated with the progression of these pathologies was also addressed. This can be observed in lines 188-192; lines 199-202; lines 278-288, and lines 358-361.

  1. The manuscript’s structure can be improved by better organizing sections to flow logically from one to the next. The current organization sometimes makes it challenging to follow the progression of ideas. For example, separating the discussion of oxidative stress, platelet alterations, and their implications for CVD into distinct sections with clear subheadings could improve readability.

R: Considering the reviewer's suggestion, subheadings were added to the section 'Oxidative stress: A "bridge" between frailty and cardiovascular diseases.' These subheadings are 'Frailty and CVD' and 'Oxidative stress in CVD and frailty.

  1. The manuscript could benefit from additional figures or tables summarizing key findings from the literature.

R: We appreciate the reviewer's suggestion. Considering this, Table 1 has been added to summarize the findings observed regarding platelet alterations in frail individuals.

  1. The analysis of how oxidative stress biomarkers can predict cardiovascular risk and the potential of antioxidant therapies lacks depth.

R: We appreciate the reviewer's suggestion. These topics were addressed in lines 278-288. This text can be seen in red font.

  1. Several sections of the manuscript contain grammatical errors and awkward phrasings that detract from the overall readability. For instance, the term "pro-thrombotic state" is used inconsistently.

R: We appreciate the reviewer's correction. The manuscript was reviewed by a grammar expert, and grammatical errors and awkward phrasings were removed.

  1. Highlighting how this review advances understanding in the field or identifies new research directions would add significant value.

R: We appreciate the reviewer's suggestion and agree with it. Considering this, a paragraph has been added to the 'Conclusion' section, which addresses new research directions and the contribution of this review. This can be seen in lines 373-378 in red font.

  1. Include more details on the methodologies of the reviewed studies to provide a clearer picture of how conclusions were reached. 
  2. We appreciate the reviewer's suggestion. Considering this, Table 1 has been added, which summarizes the main platelet alterations observed in frail patients. Additionally, the table includes a column addressing the methodology used in each of the studies.
  3. Add summary tables of key studies, their findings, and proposed mechanisms, as well as figures illustrating the relationships between oxidative stress, platelet dysfunction, and CVD.

R: We appreciate the reviewer's suggestion. Considering this, the revised manuscript presents the following points that summarize the findings obtained: a) A summary figure on the relationship between oxidative stress, platelet dysfunction, and CVD. This can be seen in Figure 1. b) A table summarizing the main findings on platelet alterations in frail individuals. This can be seen in Table 1. c) A 'Highlights' section that summarizes the main findings of the manuscript in three key points. This section can be seen after the conclusion

Reviewer 3 Report

Comments and Suggestions for Authors

Dear Editor,

I have evaluated the article.

The article makes important contributions to science. The references used are related to the subject. The article is written in a fluent language and is understandable. It contains very important results in terms of human health. However, I have some suggestions;

Too many abbreviations are used in the abstract section. Is it necessary?

Also, can the abbreviations used in the article be given as a table?

The article states that oxidative stress is a bridge between heart diseases, and this can be written in more detail.

The figure used in the article is understandable and well prepared.

Can the reference section be shortened?

Author Response

Reviewer #3

I have evaluated the article.

The article makes important contributions to science. The references used are related to the subject. The article is written in a fluent language and is understandable. It contains very important results in terms of human health. However, I have some suggestions;

  1. Too many abbreviations are used in the abstract section. Is it necessary?

R: We appreciate the reviewer's suggestion. Considering this, we have removed all abbreviations from the abstract. These changes can be observed in red font.

  1. Also, can the abbreviations used in the article be given as a table?

R: Thank you for the reviewer's suggestion. We agree with the inclusion of a supplementary table containing all the abbreviations used in the manuscript. These abbreviations can be found in Supplementary Table 1

  1. The article states that oxidative stress is a bridge between heart diseases, and this can be written in more detail.

R: Thank you for the reviewer's suggestion. In consideration of this, a paragraph addressing this topic has been added. This can be seen in lines 208-217, highlighted in red font 

  1. The figure used in the article is understandable and well-prepared.

R: We appreciate the reviewer's comments.

  1. Can the reference section be shortened?

R: We appreciate the reviewer's suggestion. In light of this, non-essential references were removed. A total of 8 references were deleted.

Round 2

Reviewer 2 Report

Comments and Suggestions for Authors

The authors have carefully addressed all the issues I raised previously. I recommend it for publication